# Towards a Unified Architecture Powering Scalable Learning Models with IoT Data Streams, Blockchain, and Open Data

Olivier Debauche [1,2,3,4,*,†] [ID], Jean Bertin Nkamla Penka [3,†] [ID], Moad Hani [3] [ID], Adriano Guttadauria [3] [ID], Rachida Ait Abdelouahid [5] [ID], Kaouther Gasmi [6] [ID], Ouafae Ben Hardouz [5] [ID], Frédéric Lebeau [2,7] [ID], Jérôme Bindelle [2,8] [ID], Hélène Soyeurt [2,4] [ID], Nicolas Gengler [2,9] [ID], Pierre Manneback [3] [ID], Mohammed Benjelloun [3] [ID] and Carlo Bertozzi [1,*] [ID]

1   Elevéo, R&D Service, Innovation Department, Awé Group, 5590 Ciney, Belgium
2   Gembloux Agro-Bio Tech, Terra, University of Liège, 5030 Gembloux, Belgium; f.lebeau@uliege.be (F.L.); jerome.bindelle@uliege.be (J.B.); hsoyeurt@uliege.be (H.S.); nicolas.gengler@uliege.be (N.G.)
3   Faculty of Engineering, ILIA Unit, University of Mons, 7000 Mons, Belgium; jeanbertin.nkamlapenka@student.umons.ac.be (J.B.N.P.); moad.hani@umons.ac.be (M.H.); adriano.guttadauria@umons.ac.be (A.G.); pierre.manneback@umons.ac.be (P.M.); mohammed.benjelloun@umons.ac.be (M.B.)
4   Gembloux Agro-Bio Tech, Modeling and Development, University of Liège, 5030 Gembloux, Belgium
5   Faculty of Sciences Ben M'sik, Hassan II University—Casablanca, Casablanca P.O. Box 7955, Morocco; rachida.aitbks@gmail.com (R.A.A.); benhardouz@yahoo.fr (O.B.H.)
6   National Engineering School of Tunis, Tunis El Manar University, 1080 Tunis, Belgium; gasmik86@gmail.com
7   Gembloux Agro-Bio Tech, Digital Energy & Agriculture Lab, University of Liège, 5030 Gembloux, Belgium
8   Gembloux Agro-Bio Tech, Animal Production Engineering and Nutrition, University of Liège, 5030 Gembloux, Belgium
9   Gembloux Agro-Bio Tech, Animal Production and Nutrition Engineering, University of Liège, 5030 Gembloux, Belgium
*   Correspondence: odebauche@awegroupe.be or olivier.debauche@uliege.be (O.D.); cbertozzi@awegroupe.be (C.B.); Tel.: +32-83-230-677 (O.D.); +32-83-230-615 (C.B.)
†   These authors contributed equally to this work.

**Abstract:** The huge amount of data produced by the Internet of Things need to be validated and curated to be prepared for the selection of relevant data in order to prototype models, train them, and serve the model. On the other side, blockchains and open data are also important data sources that need to be integrated into the proposed integrative models. It is difficult to find a sufficiently versatile and agnostic architecture based on the main machine learning frameworks that facilitate model development and allow continuous training to continuously improve them from the data streams. The paper describes the conceptualization, implementation, and testing of a new architecture that proposes a use case agnostic processing chain. The proposed architecture is mainly built around the Apache Submarine, an unified Machine Learning platform that facilitates the training and deployment of algorithms. Here, Internet of Things data are collected and formatted at the edge level. They are then processed and validated at the fog level. On the other hand, open data and blockchain data via Blockchain Access Layer are directly processed at the cloud level. Finally, the data are preprocessed to feed scalable machine learning algorithms.

**Keywords:** Internet of Things; cloud computing; fog computing; edge computing; blockchain; machine learning

## 1. Introduction

The tremendous amount of data produced by the Internet of Things (IoT ) sensors need to be collected, cleaned, verified, eventually curated, preprocessed, stored, selected, and finally explored. In parallel, it is essential to properly select relevant data from highly dynamic systems or Cyber-Physical Systems (CPS) to limit the size of the dataset to design

and train effective Machine Learning algorithms [1]. The performance of Machine Learning is classified following three criteria: quality of results, consumption of computer resources, and level of expert intervention [2]. The analysis of the dataset is composed of structured, unstructured, or semi-structured data allowing for the extraction of relevant patterns. Different kinds of data analysis can be achieved on datasets:

1. Streaming Analytics, also called event stream processing, analyzes in real time the huge volume of data. This type of analysis is adapted to detect urgent situations and immediate actions such as medical monitoring, air fleet tracking, financial transactions, traffic analysis, etc.
2. Spatial analysis analyzes geographic patterns to identify the relationships between IoT devices in the physical world. This analysis is used in applications such as smart parking.
3. Time series analysis analyzes time series data to reveal trends and associated patterns. This form of analysis is used, for example, in weather forecasting, smart meters, health monitoring systems, etc.
4. Prescriptive Analysis combines descriptive and predictive analysis to understand the best actions to take in each situation. This type of analysis is, for example, used in commercial IoT applications.

Various techniques allow for exploring, exploiting, and highlighting patterns in raw data. Figure 1 gives an overview of the different analytical approaches at the frontier of Artificial Intelligence (AI) and Data Science (DS). The AI contains inclusively Machine Learning (ML) and Deep Learning (DL). The storage in the Big Data (BD) allows for the extraction of Knowledge Discovery in Databases (KDD) by relying on machine learning.

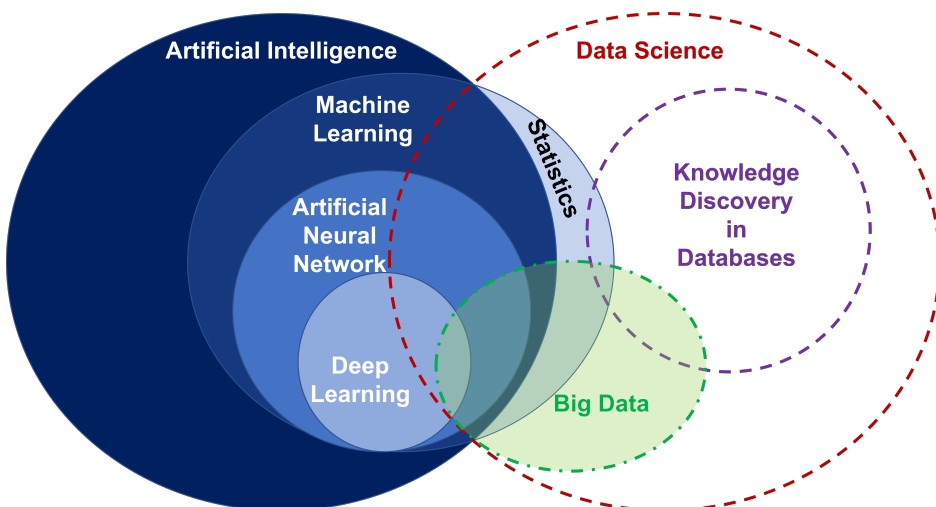

**Figure 1.** Artificial Intelligence—Data Science interactions.

Machine Learning (ML) aims to improve productivity, thus shortening the model development time to production. Machine Learning platforms exist but some suffer from difficulties that become obstacles for non-experts to use them. Among these difficulties include the complete model life cycles such as TFX [3] or even the need to transform algorithms in Kubernetes Job, for instance, MLFlow [4], or needing to integrate Application Programming Interface (API) in the code, for example in Determined [4].

Deep Learning (DL) is a sub-domain of ML in which learning is based on neural network models to model data with a high level of abstraction thanks to various non-linear transformations.

Knowledge Discovery in Databases (KDD) is an automatic discovery process from previously unknown patterns, rules, and other regular contents implicitly present in large volumes of data. KDD is often confused with Data Mining (DM), whereas DM is only one part of KDD and plays a central role in the knowledge extraction process [5].

In practical use cases, it is often necessary to combine disparate data sources to construct a comprehensive training database for machine learning algorithms. However, these data sources exhibit variations in terms of quality and data formats. To ensure consistency across these sources, preprocessing steps are required to consolidate the training database. Furthermore, the arrival frequencies, speeds, and quantities of data can greatly differ depending on the systems and devices responsible for their production [6].

To address these challenges, Lambda and Kappa architectures are commonly employed for real-time or batch data processing [7,8]. These architectures provide robust frameworks for managing and processing data in different scenarios. The Lambda architecture allows for the simultaneous processing of both real-time and batch data, enabling near-real-time analytics while ensuring fault tolerance and scalability. On the other hand, the Kappa architecture is primarily focused on real-time data processing, leveraging stream processing frameworks to handle high-velocity data streams efficiently.

By leveraging Lambda or Kappa architectures, organizations can effectively handle the complexities associated with combining diverse data sources. These architectures facilitate the integration of data with varying formats, qualities, and arrival frequencies, allowing for the creation of a unified and consistent training database. Through appropriate preprocessing steps and the utilization of the appropriate architecture, organizations can unlock valuable insights and train machine learning algorithms on comprehensive and reliable datasets.

To illustrate the need for an integration of consolidation of multiple sources and machine learning platforms, four use cases are provided:

1. Internet of Things (IoT) Predictive Maintenance: IoT devices generate a massive volume of data streams, which vary in arrival frequency, speeds, and quantities. In the context of predictive maintenance, combining these heterogeneous data sources is crucial for building accurate machine-learning models. Data from sensors, machine logs, maintenance records, and external sources need to be integrated and preprocessed to create a comprehensive training database.
2. Fraud Detection in Financial Transactions: In the realm of financial transactions, fraud detection is a critical use case that requires combining diverse data sources to build a reliable training database. Various sources such as transaction logs, customer profiles, and external data feeds contribute to the training data. However, these sources often differ in data formats and quality. To ensure consistency, preprocessing steps such as data cleaning, normalization, and feature engineering are applied to consolidate the training database.
3. Crop Yield Optimization in Smart Farming: In smart farming, optimizing crop yield and ensuring efficient resource utilization is crucial for sustainable and profitable agricultural practices. To achieve this, farmers often gather data from various sources such as soil sensors, weather stations, crop health monitors, and machinery telemetry. However, these data sources may have different quality levels, formats, and update frequencies. This is where the Consolidated Learning (CL) platform comes into play, facilitating the preprocessing and integration of data for effective decision making.
4. Threat Detection and Response in Cybersecurity: In the field of cybersecurity, organizations face constant challenges in detecting and responding to evolving cyber threats. To effectively protect their systems and data, organizations rely on various security tools and systems that generate vast amounts of security event logs, network traffic data, and intrusion detection alerts. However, these data sources often differ in terms of data formats, protocols, and logging mechanisms, making it difficult to analyze and correlate them for accurate threat detection.

This paper aims to present a simple and agnostic Machine Learning platform to combine and process the Internet of Things data, blockchains data, and data stream data, managing the entire model life cycle while allowing users to deploy their models on their usual frameworks: TensorFlow (tensorflow.org; accessed on 11 November 2022), Apache MXNet [9], Pytorch [10], Keras (keras.io; accessed on 15 September 2022), and Horovod [11].

The contributions in this paper are:

1. An Integrated Architecture Edge-Fog-Cloud and Machine Learning;
2. Complete automation of the data acquisition chain, data reconciliation, development, and training of machine learning models.

The rest of this paper is organized as follows. Section 2 summarizes recent advances in ML platforms. In Section 3, the architecture is conceptualized. In Section 4, the architecture is described. In Section 5, the architecture is implemented, and in Section 6, experimentation is achieved to demonstrate its abilities and performances. Then, the obtained results are analyzed in Section 7. Afterward, in Section 8, the limitations of this work are discussed. Finally, this work is concluded and future research directions are drawn in Section 9.

## 2. Related Works

In this section, the recent advances published on the ML platform are summarized. Afterward, we compare the pros and cons of each one to identify the issues to address using the proposed architecture, thus also highlighting the contributions of this paper.

Lee et al. [2] proposed an autonomic ML platform based on five autonomic levels using steps according to the degree of expert intervention. The proposed platform can operate as a black box for non-expert users without writing code and as a white box for expert users where they can write code to design ML tasks using script programming language [2].

Bagozi et al. [1] presented the Interactive Data Exploration As-a-Service (IDEAaS) approach compliant with the Human-In-The-Loop Data Analysis (HILDA) vision, enabling Big Data Exploration (BDE) and including an incremental clustering algorithm, a multi-dimensional organization of summarized data, and data relevance evaluation techniques. IDEAaS architecture is organized in three modules: *Data collection*, which achieves the preprocessing of IoT collected data stored in the form of JavaScript Object Notation (JSON) document in MongoDB; *Core*, which summarizes a batch of collected data and generates a synthesis and snapshots organized in JSON at each iteration; and *Front-end*, which takes the form of a data exploration GUI that loads summarized data and snapshots and then activates the data exploration API [1].

Apache Submarine [12] is a unified ML platform that takes into account not only expert data scientists but also citizen data scientists. This platform allows fitting models with a large parameter space and/or with a high computational cost thanks to the support of model parallelism to shorten the learning process. Regarding Submarine support, on one side, Kubernetes is the most popular container orchestrator, and on the other side, Apache Hadoop YARN can be easily integrated into the Hadoop ecosystem. Submarine integrates also Azkaban (azkaban.github.io; accessed on 22 October 2022) which is useful to submit a tasks set of workflow in Apache Spark for preprocessing or transfer Apache Zeppelin (zeppelin.apache.org; accessed on 10 September 2022) to Azkaban to distribute Deep Learning on Tensorflow (tensorflow.org; accessed on 10 September 2022).

The data stream clustering, *CluStream*, has emerged as a solution to process data arriving at a high rate and synthesize them by a lossless aggregation dividing the clustering into an online component that periodically stores detailed summary statistics and an offline component that uses these summary statistics to provide a quick understanding of map clusters in the data stream [13]. This approach has undergone various improvements and adaptations, such as the use of sliding windows [14,15]; Equi-Clustream, a dynamic clustering of mixed-type time-evolving data [16]; Clustream, which is a Spark implementation [17]; and speed up [18,19].

Akbar et al. [20] presented a generic architecture that combined ML with Complex Event Processing (CEP) to predict complex events for proactive IoT applications. They proposed an adaptive prediction algorithm called Adaptive Moving Window Regression (AMWR). This algorithm utilizes a moving window of data for training the model, and when new data arrives, it calculates an error and retrains the model based on it. This

approach is interesting because it re-trains the model as soon as possible to obtain the potential of the new data.

Calabrese et al. [21] designed an event-based IoT and ML architecture for Predictive Maintenance in Industry 4.0. This architecture is divided into three modules: the data acquisition module built with Azure Blog Storage; the data processing defined with Apache Spark and the data science module based on the pySpark libraries; and the predictive monitoring web application module. The final predictions result in two data processing types: the offline mode to generate predictive models based on historical log data files and the online mode which uses the last 24 h data. This approach is interesting because it is similar to the lambda architecture but with ML tools.

Machorro-Cano et al. [22] presented a big data and machine learning-based smart home energy management system for home comfort, safety, and energy saving, called HEMS-IoT. Its architecture is defined by seven layers: the Device layer (sensors, actuators), the Communication layer (TCP/IP, HTTP/IP), the Data layer (sensed data, recommendations), the Management layer (recommender system, user management), the IoT services layer (REST API), the Security layer (authentication, authorization), and the Presentation layer (web and mobile application). This architecture proposes many detailed layers based on their choice, but they can be presented a little differently.

Rashid et al. [23] developed a smart energy monitoring system based on intelligence in IoT. This architecture is defined by a common three-layer view: the physical sensing layer, the IoT middle layer, or the network layer where the ML model is trained on Google Colab, and the application layer. This architecture is simple and focuses on the integration of the ML model to predict energy consumption.

Elsisi et al. [24] developed an integrated IoT architecture based on Deep Neural Network (DNN) to handle the problems of cyber-attacks on Automated Guided Vehicles (AGV). In addition to the usual tiers of real-time IoT architecture, in this architecture, an additional unit was coupled between the sensors and the IoT platform. Based on the proposed DNN against cyber-attacks, this unit aims to analyze the transferred data and prevent known cyber-attacks.

Flores-Martin et al. [25] developed an architecture based on IoT and ML to improve the monitoring of elderly people in nursing homes. This architecture has three parts: the inputs (the collected data), the controller (including the ML model), and the outputs (the actuators).

Anh Khoa et al. [26] proposed smart trash bins with real-time monitoring. This architecture is defined by the trash bin with a sensor, the intermediate server which collects the data sent through the LoRa gateway, the cloud server, and finally, an application to display the data.

Table 1 summarizes the pros and cons of related work.

**Table 1.** Related works summary.

| References | Pros | Cons |
|---|---|---|
| Lee et al. [2] | Minimize expert intervention | Focus on the cloud only |
| Bagozi et al. [1] | New approach to use data as a service | Complex and too theoretical |
| Apache Submarine [12] | Open for non-experts of ML | Difficult to install for non-IT Experts |
| Akbar et al. [20] | The AMWR algo to retrain the model with new data | Generic architecture |
| Calabrese et al. [21] | The idea of Lambda architecture with ML tools | More complex to maintain like Lambda |
| Machorro-Cano et al. [22] | Detailed layers | ML part is not clearly explained |
| Rashid et al. [23] | Detailed layers based on LSTM | Focuses on the network layer with ML |
| Elsisi et al. [24] | Used a DNN model against cyber-attacks | Implemented DNN is not clearly explained |
| Flores-Martin et al. [25] | Used ML for monitoring elderly people | The ML model tiers are partially defined |
| Anh Khoa et al. [26] | Lots of details about the materials | Lack of details on the ML model |

## 3. Architecture Conceptualization

An automatic ML platform must provide, on one hand, ML frameworks and tools such as SparML (spark.apache.org/docs/latest/ml-guide.html; accessed on 16 October 2022), TensorFlow (tensorflow.org; accessed on 10 September 2022), Keras (keras.io; accessed on 15 September 2022), Pytorch (pytorch.org; accessed on 10 September 2022), Apache Mahout (mahout.apache.org; accessed on 16 October 2022), Weka (weka.io; accessed on 16 October 2022), and so forth, and language environments such as SparkML, Python, and so on for experts. On the other hand, the platform must allow non-experts to be able to build ML applications by the automation of tasks. In addition, the platform must be able to execute training and inference tasks on distributed and parallel computing resources.

To enable the development of machine learning models, data must first be combined, processed, and verified before constituting a training database. The data can come from different sources in different forms: continuous data stream, intermittent to transmitted at regular intervals, or data bursts. The data are rarely in the same format and must therefore be transformed to be standardized or adapted. Furthermore, the quality of the data is also variable depending on the data source. For all these reasons, a data preprocessing and processing chain must be set up before the machine learning modeling platform.

The conceptual architecture is composed of three layers (see Figure 2).

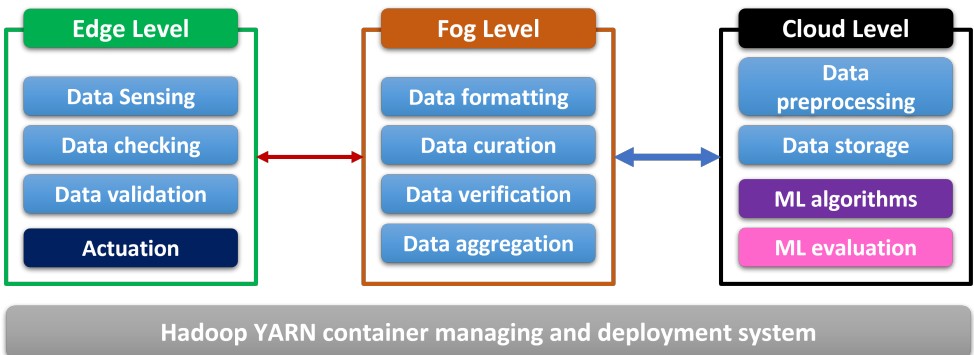

**Figure 2.** Conceptual diagram of the architecture.

It is organized in levels from the bottom to the top as follows (Left to Right on the Figure 2):

- An Edge layer (EL) is composed of microcontrollers where data are acquired using sensors and then checked and validated. EL is also responsible for the activation of the actuators that allow for acting on the environment of the subject of observation or the subject himself. See the left part of Figure 2. The EL operates at the edge of the network, closer to the data source, which offers several advantages. By processing data locally, the EL reduces latency and minimizes dependence on cloud-based services, making it well-suited for time-critical applications. Moreover, the EL enhances data security and privacy since sensitive information can be processed and analyzed locally, without necessarily transmitting it to external servers.
- A Fog level (FL) is the gateway that ensures network protocols commutation. At the same time, the data are centralized for verification and curated where possible before being transmitted to the cloud. See the middle section of Figure 2. The FL acts as a communication bridge, ensuring seamless connectivity between different network protocols utilized by the EL and the Cloud layer. It enables the exchange of data and commands, allowing for efficient and reliable communication across the entire system. By providing protocol commutation capabilities, the FL overcomes the challenges posed by heterogeneous network environments where various devices and protocols coexist.
- A Cloud level (CL) is where IoT data are processed and stored in a database as a time series. See the right part of Figure 2. In the CL, data received from the Fog

level (FL) or other sources are processed and analyzed using various cloud-based services, algorithms, and machine learning models. This processing stage involves extracting valuable insights, detecting patterns, and generating actionable information from the collected IoT data. The CL leverages the computational resources and scalability of cloud platforms to handle large volumes of data and perform complex computations efficiently.

## 4. Architectural Proposal

The agnostic and unified architecture described in the previous section encompasses the entire chain of data acquisition up to the Machine Learning model. This architecture is designed to efficiently process and prepare data, enabling its seamless utilization in various Machine Learning algorithms. Here, we provide a detailed description of this architecture. The architecture begins with data acquisition, which involves collecting data from diverse sources such as sensors, devices, or external systems. The Edge layer (EL) plays a vital role in this process, utilizing microcontrollers and sensors to acquire data. The EL ensures that the acquired data are checked and validated before being passed to the next layer. Once the data are acquired by the EL, it is forwarded to the Fog level (FL). The FL acts as a gateway, facilitating network protocol commutation and centralizing the data for verification and curation. It ensures that the data are valid and reliable before it is transmitted to the Cloud level (CL) for further processing. In the CL, the data are processed using cloud-based services, algorithms, and machine learning models. This processing stage involves analyzing the data, extracting insights, and preparing it for utilization in Machine Learning algorithms. Various techniques such as data cleaning, normalization, feature extraction, and dimensionality reduction may be applied to efficiently prepare the data for modeling. The processed data are stored in a database, typically organized as a time series, in the CL. This database enables efficient storage, retrieval, and querying of the data, maintaining its chronological order. The stored data serve as a valuable resource for historical analysis, trend identification, and generating reports or visualizations. To ensure flexibility and adaptability, the architecture is designed to be agnostic, allowing it to work with different types of data and accommodate various Machine Learning algorithms. It supports the integration of different data formats, protocols, and sources, ensuring seamless compatibility across the system. The unified nature of the architecture ensures that all stages of the data chain, from acquisition to the Machine Learning model, are managed within a cohesive framework. This eliminates the need for fragmented and disparate systems, streamlining the entire process and enabling efficient data flow. By efficiently processing and preparing data, the architecture enables easy utilization of the data in diverse Machine Learning algorithms. This versatility empowers organizations to leverage the data for various purposes, such as predictive modeling, anomaly detection, or decision support systems. In summary, the agnostic and unified architecture described above provides a detailed framework for managing the complete data chain, from acquisition to the utilization of Machine Learning algorithms. It emphasizes efficient data processing and preparation, ensuring compatibility with different data types and Machine Learning techniques. By adopting this architecture, organizations can harness the full potential of their data and drive insights and value from it. Figure 3 shows an overview of the architecture.

### 4.1. Edge Layer

The EL is responsible for the sensing and actuating. This level is composed of ARM microcontrollers using TrustZone [27] in order to obtain security through hardware-based isolation. It is important to separate important data such as private keys, user data, and security functions from generic data and functions such as GUI elements or the real-time operating system (RTOS). To achieve this, a hardware mechanism called TrustZone is implemented in single-core microcontrollers. TrustZone divides the execution environment into secure and non-secure memory, peripherals, and functions. Additionally, each execution environment includes a memory protection unit (MPU) that can be utilized to further

isolate memory regions. This added layer of isolation can deter potential attackers from attempting to access data. TrustZone and MPU provide a multi-layered security approach to safeguard important data. At this level, devices are constrained in terms of memory, storage, and processing capabilities. As a consequence, the abilities to achieve complex treatments are strictly limited.

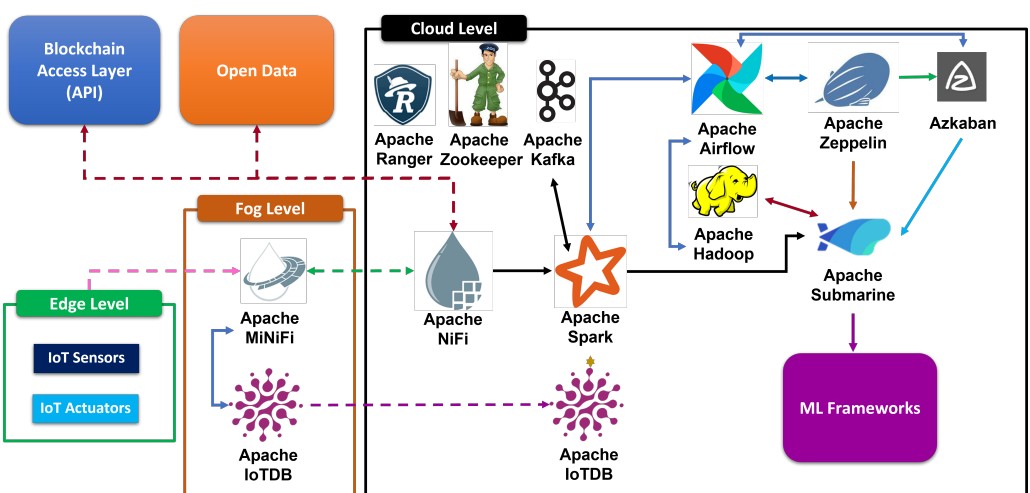

**Figure 3.** Architecture Software Components Overview.

### 4.2. Fog Layer

The FL, also known as the edge-cloud layer, serves as an intermediate level between the EL and the CL. The primary function of the FL is to carry out more extensive treatments compared to the other two layers. These treatments may include data processing, analysis, and storage, among others. By processing data at the edge, the FL reduces the latency associated with transmitting data to the Cloud Layer for processing. This allows for faster decision making and improved overall system performance.

In addition to data processing, the FL is also responsible for protocol conversion to enhance the transmission of data between wireless sensor networks and Internet protocols. This is achieved through the use of protocols such as MQ Telemetry Transport (MQTT), Constrained Application Protocol (CoAP), and Hypertext Transfer Protocol (HTTP), which allows for seamless communication between devices. The FL plays a crucial role in the performance and efficiency of the IoT by ensuring that data are properly processed and transmitted. By optimizing data transmission, the FL enables IoT systems to operate at peak efficiency while conserving energy and reducing network congestion. Overall, the FL acts as a bridge between the EL and CL, bringing the benefits of both worlds together to create a powerful and efficient IoT ecosystem.

Apache MiNiFi (nifi.apache.org/minifi; accessed on 10 September 2022) is a complementary approach to data collection that complements Apache NiFi's (nifi.apache.org; accessed on 10 September 2022) core principles of data flow management by focusing on collecting data at the source of its creation. Apache IoTDB (iotdb.apache.org; accessed on 10 September 2022) is a native lightweight and high-performance database for IoT, deployable on the edge and the cloud.

### 4.3. Cloud Layer

The CL consists of server infrastructure that incorporates Intel CPUs featuring Software Guard Extensions (SGX) technology [27]. This innovative technology ensures secure data processing by isolating sensitive computations within a protected memory enclave. To support its operations, the CL relies heavily on Apache Submarine [12], an integrated Machine Learning platform that comprises three key components: a user interface, a server module, and a container management, orchestration, and deployment system.

The user interface of Apache Submarine facilitates a seamless and intuitive experience for users, allowing them to interact with the CL's functionalities effortlessly. Through this interface, users can manage and monitor their machine learning tasks, access datasets, and configure training parameters, ensuring a streamlined workflow.

At the core of the CL lies the server module of Apache Submarine. This module is responsible for handling the intricate processing and computation tasks required by machine learning algorithms. Leveraging the power of Intel CPUs equipped with SGX technology, the server module ensures that sensitive data are processed securely and efficiently within isolated enclaves, protecting it from unauthorized access or tampering.

To ensure efficient resource utilization and scalability, the CL incorporates a container management, orchestration, and deployment system. This system efficiently manages the deployment and execution of machine learning tasks within a distributed environment, allowing for the seamless allocation of resources and optimizing performance. By leveraging containerization technologies such as Docker or Kubernetes, the CL can dynamically scale its computational resources based on the workload, enabling the processing of large-scale machine learning tasks with ease.

Apache Hadoop YARN plays the role of a fine-grained Graphics Processing Unit (GPU) scheduler and allows for the scheduling of more than 1000 containers per second. It was preferred to Kubernetes, which is only able to schedule 100 containers per second because of the data storage in the etcd (etcd.ioits; accessed on 22 October 2022) database, which induces a significant latency and limits performance (submarine.apache.org; accessed on 10 September 2022).

In summary, the CL harnesses the power of Intel CPUs with SGX technology and relies on Apache Submarine's comprehensive platform to provide a secure and efficient environment for machine learning tasks. Through its user interface, server module, and container management system, the CL offers a cohesive ecosystem that empowers users to leverage advanced machine learning capabilities while ensuring data confidentiality and integrity.

Apache Zookeeper is a centralized service used for preserving configuration details, naming, giving out distributed synchronization, and offering group services (zookeeper.apache.org; accessed on 10 September 2022). Apache NiFi (nifi.apache.org; accessed on 10 September 2022) is a powerful, scalable, and distributed directed graph of data routing, transformation, and system mediation logic. Apache IoTDB [28] is deeply integrated with Apache Spark and Apache Hadoop, allows to address requirements of massive data storage with ingestion at high speed, and provides abilities for complex data analysis. Apache Spark (spark.apache.org; accessed on 10 September 2022) is an open-source unified framework for distributed computing using implicit data parallelism and fault tolerance. Apache Airflow (airflow.apache.org; accessed on 10 September 2022) is an open-source management platform designed to schedule and monitor workflows. Apache Zeppelin (zeppelin.apache.org; accessed on 10 September 2022) is a web-based notebook that facilitates interactive data analytics, data-driven collaboration, and the creation of collaborative documents. Azkaban (azkaban.github.io; accessed on 22 October 2022) is a batch workflow job scheduler that supports the composition of workflows, which can include multiple batch processing activities and run Hadoop jobs. Apache Ranger (ranger.apache.org; accessed on 10 September 2022) is a centralized platform for defining, administering, and globally managing security policies for Hadoop clusters. Apache Kafka (kafka.apache.org; accessed on 10 September 2022) is an open-source distributed event streaming platform that provides a unified, real-time, low-latency system for handling data streams used for high-performance data pipelines, and streaming analytics. Apache Hadoop (hadoop.apache.org; accessed on 10 September 2022) is a platform that permits the distributed handling of vast datasets across computer clusters by using straightforward programming models. Apache Submarine (submarine.apache.org; accessed on 10 September 2022) is a comprehensive ML platform designed to enable data scientists to develop complete Machine Learning workflows. Submarine can train models based on TensorFlow, PyTorch, MXNet, or by

using the framework Horovod (horovod.ai; accessed on 10 September 2022), a distributed deep-learning training framework that allows for deploying in multi-nodes and multiGPU models built with MXNet, Keras, PyTorch, or TensorFlow. TensorFlow [29] is an end-to-end open-source platform for machine learning developed by Google. PyTorch [10] is an open-source machine learning framework that accelerates the path from research prototyping to production deployment. Apache MXNet (mxnet.apache.org; accessed on 10 September 2022) is an open-source flexible and efficient library for deep learning used to train and deploy deep neural networks. Keras (keras.io; accessed on 15 September 2022) is an API that has been specifically designed to minimize the number of user actions required for common use cases while also providing clear and actionable error messages. Figure 4 below shows ML Frameworks usable with Submarine.

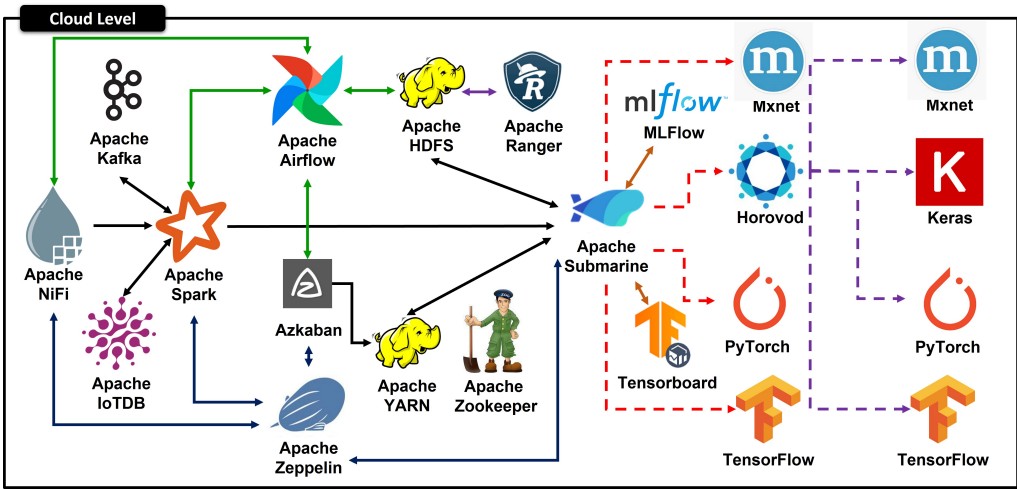

**Figure 4.** Components of the cloud part of the architecture.

### 4.4. Security

At the cloud level, the security at the Apache Hadoop cluster is globally managed by Apache Ranger, which centralizes access policies to files, folders, databases, tables, or even columns. These policies can apply to individual users as well as groups. Ranger Key Management Service (RKMS) provides a scalable encryption key management service that complements the native function of Hadoop Key Management Service (HKMS) by allowing keys to be stored in a secure database on the one hand and encrypting Hadoop Distributed File System (HDFS) "data at rest" on the other. Ranger provides a centralized auditing capability that monitors all access requests in real time and supports multiple destination sources such as Apache Hadoop HDFS, Apache Kafka, Apache NiFi, YARN, Elasticsearch, etc. The Ranger Audit component collects and then stores ranger audit logs in Elasticsearch ( elastic.co/elasticsearch; accessed 18 September 2022), and shows logs for each access event of the resource in the Ranger Audit User Interface (UI).

### 4.5. Software Components

In this paragraph, we summarize the identified software components that will be used to implement the architecture proposal in Section 5. Table 2 summarizes tools and software used to build the architecture.

**Table 2.** Tools and their role in our architecture.

| Tools | Usage | Role |
|---|---|---|
| Apache MiNiFi | EL | Collects data at the source of its creation. |
| Apache NiFi | FL | Routes and transforms data on the desirable format. |
| Apache IoTDB | FL, CL | Addresses requirements of massive data storage with ingestion at high speed. |

**Table 2.** *Cont.*

| Tools | Usage | Role |
|---|---|---|
| Apache Zookeeper | CL | Centralizes configuration details, and names and gives out distributed synchronization. |
| Apache Spark | CL | Distributes computing using implicit data parallelism and fault tolerance. |
| Apache Hadoop | CL | Permits the distributed handling of vast datasets across Computer clusters. |
| Apache Airflow | CL | Schedules and monitor workflows. |
| Apache Zeppelin | CL | Facilitates interactive data analytics. |
| Azkaban | CL | A batch workflow job scheduler that supports the composition of workflows. |
| Apache Ranger | CL | Defines, administers, and manages security policies for Hadoop clusters. |
| Apache Kafka | CL | Provides a unified, real-time, low-latency system for handling data streams. |
| Apache Submarine | CL | Helps to develop ML workflows. |
| TensorFlow | CL | Helps to develop ML model. |
| PyTorch | CL | Accelerates the path from research prototyping to production deployment. |
| Apache MXNet | CL | Uses for Deep Learning used to train and deploy deep neural networks. |
| Keras | CL | Helps to minimize the number of user actions required for common use cases. |
| Horovod | CL | A distributed deep-learning framework. |

## 5. Implementation

To deploy the proposed architecture, Docker containerization technology is selected because it is widely used in the DevOps context to quickly prototype architectures. In the following, we detail the content of the different services at the cloud level and then at the edge and fog levels. Figure 4 shows the interactions between services composing the cloud level of our architecture.

For the cloud part of our architecture, Docker containerization is used. The list of software components that composed each service is resumed in the tables below.

Table 3 presents the three services that are deployed in the Kafka service, which temporarily stores data before its processing. Zookeeper ensures the coordination of the service and uses server ports 2888 and 3888 while the client port 2181 is used with the instance of Kafka to communicate with it. In addition, the Kafka instance listens on two different external ports: 9092 and 29,092, and on the internal port: 19,092. Kafka Schema Registry listens to requests on port 8081.

**Table 3.** Composition of Kafka Component.

| Service | Docker Image | Instance | Internal Port | Exposed Port | Mapped Port |
|---|---|---|---|---|---|
| Zookeeper | confluentinc/cp-zookeeper:7.2.1 | 1 | 2888, 3888 | 2181 | 2181 |
| Kafka | confluentinc/cp-kafka:7.2.1 | 1 | 19,092 | 9092, 29,092 | 9092, 29,092 |
| Kafka Schema Registry | confluentinc/cp-schema-registry:7.2.1 | 1 | | 8081 | 8081 |

Apache NiFi 1.20.0 is installed in cluster mode with Zookeeper 3.8.0 and Apache NiFi Registry 1.20.0 to make persistently developed workflows. To avoid conflict, the client port of NiFi 8080 was mapped with exposed port 8091 while the NiFi registry client port was conserved on 18,000. Apache NiFi is linked with ZooKeeper with its default client port: 2181, while port 8082 is reserved for communication with NiFi nodes. The composition of the services is summarized in Table 4. Apache NiFi uses the processor *PutIoTDB* to read the content of the incoming *FlowFile* from NiFi and classifies them as individual records before writing them in Apache IoTDB 0.13.3 using the native interface.

**Table 4.** Composition of the NiFi Component.

| Service | Docker Image | Instance | Internal Port | Exposed Port | Mapped Port |
|---|---|---|---|---|---|
| Zookeeper | bitnami/zookeeper:3.8.0 | 1 | 8082 | 2181 | 2181 |
| NiFi | apache/nifi:1.20.0 | 1 | 8082 | 8080 | 8091 |
| NiFi Registry | apache/nifi-registry:1.20.0 | 1 | | 18,000 | 18,000 |

Apache Zeppelin version 0.10.1 was built from a personalized Dockerfile to add the connection to Apache IoTDB version 0.13.3 using its interpreter compiled from the source code of Apache IoTDB with Apache Maven 3.8.6 and OpenJDK 19. The default port 8080 was mapped with the exposed port: 5000, as illustrated in Table 5. Zeppelin was also connected with Apache Spark 3.2.2 with its native interpreter allowing experts to adapt processing pipelines directly. While Apache Spark and Apache IoTDB were linked using the *spark-iotdb-connector* compiled in Scala 2.12.

**Table 5.** Composition of the Zeppelin Component.

| Service | Docker Image | Instance | Internal Port | Exposed Port | Mapped Port |
|---------|--------------|----------|---------------|--------------|-------------|
| Zeppelin | Custom Dockerfile | 1 | | 8080 | 5000 |

To evaluate Apache Submarine quickly, we opted to use the mini-submarine proposed in the dev-support directory of the Apache Submarine repository available on GitHub. We compiled Apache Submarine source code using Maven 3.6.3 and Java 8. Afterward, an image based on the Dockerfile which stacks Apache Hadoop 2.9.2, Apache Zookeeper 3.4.14, Apache Spark 3.4.3, and the SNAPSHOT of the GitHub master branch of Apache Submarine was built.

Table 6 describes services implemented in the Airflow component. The *Airflow-init service* is triggered at the service's launch and exits after execution. Airflow-webserver and Airflow-scheduler are responsible for the front end of the service and scheduling of workflow respectively. The scheduler exposes port 8793 and the server exposes port 8000 mapped with the default port 8080. All the data of the service are stored in Postgresql 14 which listened on port 5434 mapped with the default port 5432.

**Table 6.** Composition of the Airflow Component.

| Service | Docker Image | Instance | Internal Port | Exposed Port | Mapped Port |
|---------|--------------|----------|---------------|--------------|-------------|
| Airflow-scheduler | apache/airflow:2.6.0 | 1 | | 8793 | 8793 |
| Airflow-webserver | apache/airflow:2.6.0 | 1 | | 8080 | 8080 |
| Airflow-init | apache/airflow:2.6.0 | 1 | | | |
| Postgresql | postgres:14 | 1 | | 5432 | 5434 |

To reduce the deployment time of the Docker images, a registry coupled with Redis allows the storage of local Docker images built by researchers. Table 7 shows the composition of the service.

**Table 7.** Composition of the Registry Component.

| Service | Docker Image | Instance | Internal Port | Exposed Port | Mapped Port |
|---------|--------------|----------|---------------|--------------|-------------|
| Docker Registry | registry:2.8.2 | 1 | | 5000 | 5000 |
| Redis | bitnami/redis:7.0.5 | 1 | | 6379 | 6379 |

The Fog level within the architecture plays a vital role in establishing a seamless connection between microcontrollers embedded in connected devices, which are equipped with sensors and/or actuators, and the Cloud level. To ensure optimal resource utilization, a lightweight variant of Apache NiFi, known as Apache MiNiFi, is deployed on the microcontrollers. This streamlined version, coupled with IoTDB, consumes fewer resources while maintaining efficient performance. The Fog IoTDB instance synchronizes seamlessly with the cloud-based IoTDB using TsFile sync, enabling smooth data transmission and synchronization. Apache MiNiFi serves as a versatile component, capable of receiving data from Edge devices through various Radio Frequency protocols. Upon receiving the data, Apache MiNiFi processes it using its built-in capabilities before routing it through the *PutIoTDB* processor. To facilitate seamless integration, the 'Record Reader' configuration

reads the content of the incoming FlowFile as separate records, ensuring efficient handling and transmission to Apache IoTDB through its native interface.

## 6. Experimentation

To validate the proposed architecture, two experiments were conducted: In the first, a simple classifier was tested on open data. In the second, data from the Ethereum blockchain were retrieved via Access Layer API, and the same classifier was trained. Experimentation was conducted on a VPS Contabo with 10 vCPU Cores, 60 GB RAM, and 800 GB NVMe (contabo.com/en/vps/; accessed on 16 October 2022).

### 6.1. Description of the Dataset

The dataset MNIST (nist.gov/srd/nist-special-database-19; accessed on 15 January 2023) is composed of 70,000 images of 28 × 28 px divided into ten classes as follows: class 0: 6903 data; class 1: 7877 data; class 2: 6990 data; class 3: 7141 data; class 4: 6824 data; class 5: 6313 data; class 6: 6876 data; class 7: 7293 data; class 8: 6825; class 9: 6958 data. Figure 5 shows the distribution of data in the 10 different classes.

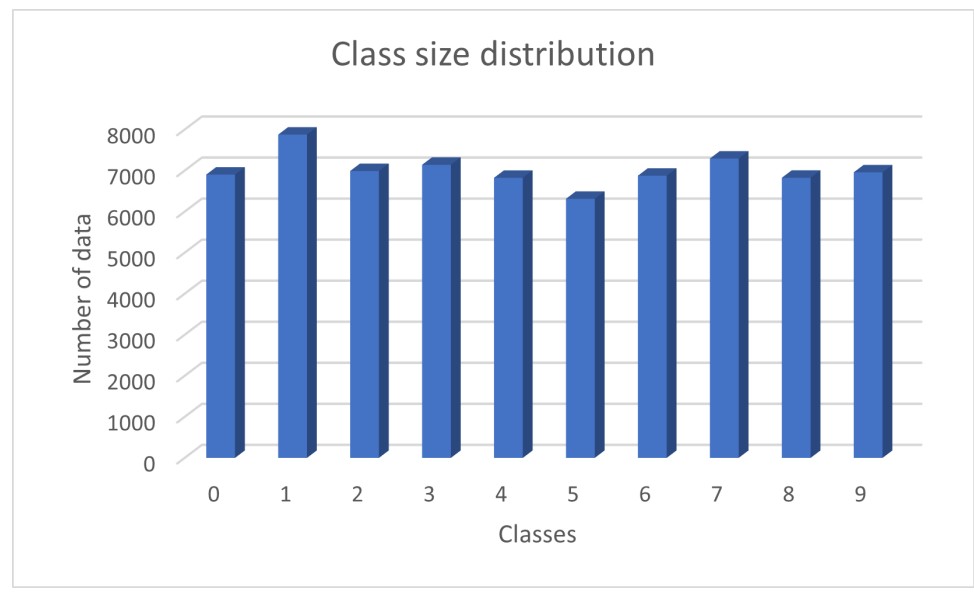

**Figure 5.** Class size distribution.

### 6.2. Structure of the Classifier

The classifier that is used to evaluate our architecture is composed of 8 layers from the bottom to the top:

1.  Convolutional Layer 1 (Conv2D):
    - Filters: 32;
    - Kernel size: (3, 3);
    - Activation function: ReLU;
    - Input shape: (28, 28, 1) (images with a size of 28 × 28 and a single channel).
2.  Max Pooling Layer 1 (MaxPooling2D):
    - Pool size: (2, 2).
3.  Convolutional Layer 2 (Conv2D):
    - Filters: 64;
    - Kernel size: (3, 3);
    - Activation function: ReLU.
4.  Max Pooling Layer 2 (MaxPooling2D):
    - Pool size: (2, 2).

5. Convolutional Layer 3 (Conv2D):
   - Filters: 64;
   - Kernel size: (3, 3);
   - Activation function: ReLU.

6. Flatten Layer:
   - Converts the 2D output from the previous layer into a 1D vector to feed into a dense layer.

7. Dense Layer 1 (Dense):
   - Neurons: 64;
   - Activation function: ReLU.

8. Dense Layer 2 (Dense):
   - Neurons: 10;
   - Activation function: Softmax;
   - Output layer with 10 neurons, representing the probability distribution over the 10 classes.

Figure 6 presents the organization of the 8 layers of the classifier.

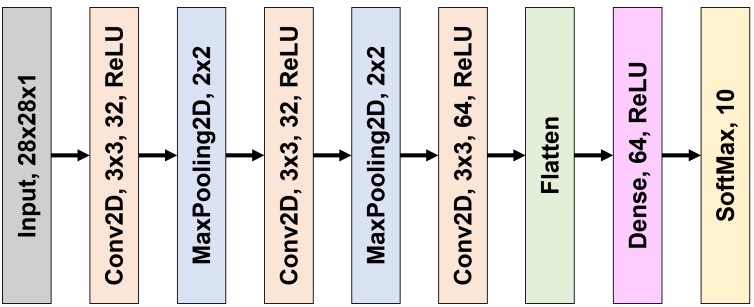

**Figure 6.** Structure of the classifier.

The model was deployed using TensorFlow 2.2.

### 6.3. Training of the Model

The model was trained on 10 epochs with 70 steps per epoch with the Adam optimizer loss function: sparse categorical cross-entropy, and the Adam Optimizer on one side with open data and on the other side with data from the blockchain on a single CPU with a RAM limited to 2G. Afterward, the accuracy of the model was evaluated with training using different learning rates and batch sizes, and configurations with one or two CPUs.

## 7. Results and Discussion

The first experiment involves testing the use of open-source data by loading MNIST data directly from a ZIP file hosted on a remote server. The second experiment involves retrieving MNIST data previously stored in the Ethereum blockchain and then retrieving it using the Access Layer APIs. This data are then prepared using Apache Spark at the cloud level so that it can be used for training on Apache Submarine.

The experiments have shown that the cloud architecture can work with data from files with open data and data streams from the Ethereum blockchain. Figures 7 and 8 show results obtained from files (open data) and stream data (blockchain), respectively.

Figures 7 and 8 give similar results in terms of accuracy and loss, but the training speed is very different between the two experiments. Training with data from the blockchain is 5 times slower than reading data from a file. This difference in speed is explained by the time it takes to access the data stored in the blockchain via the access layer API and transform it into the cloud to prepare it for training.

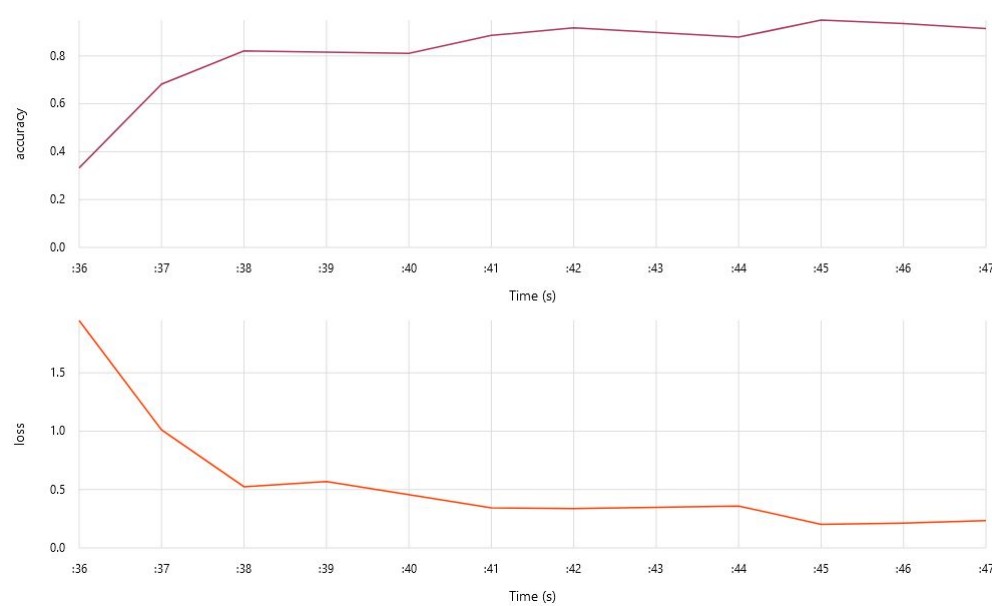

**Figure 7.** Evolution over time of the accuracy and the loss during the model training with open data.

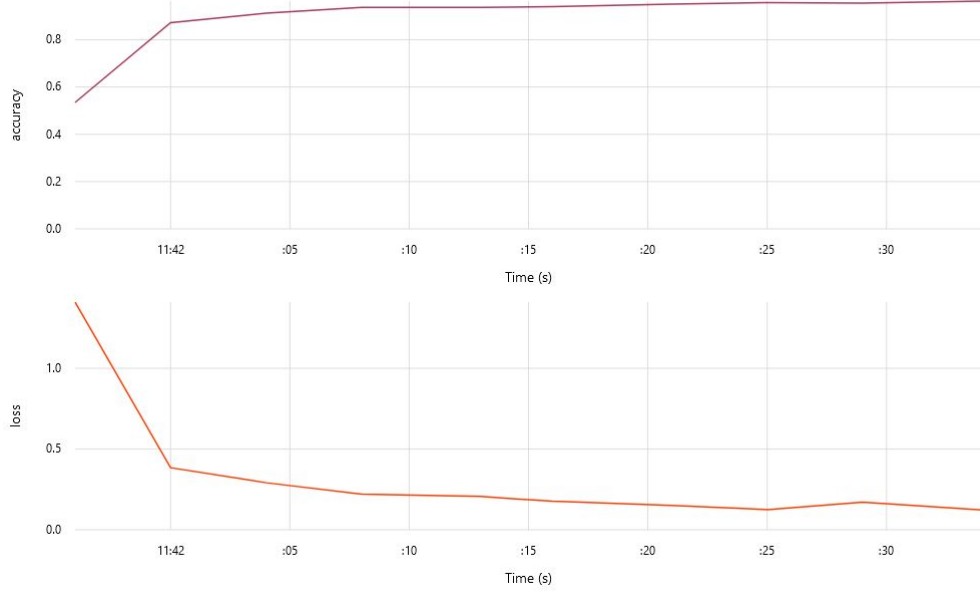

**Figure 8.** Evolution of the accuracy and the loss during the model training with blockchain data.

Table 8 presents accuracies obtained for different configurations of the learning rate, batch size, number of CPU, and RAM. The analysis of the table shows that better results are obtained with a learning rate (lr) of 0.001 and a batch size of 150. The accuracy increases progressively with the batch size up to maximum with 150 images. In addition, the distribution of the training does not impact the accuracy but reduces significantly the processing time.

A licensing analysis of the proposed architecture was achieved. Table 9 includes the licenses for the software components used in the architecture. The table shows that all the architecture is open source with licenses which allow for contributions without license contamination of the code.

**Table 8.** Accuracy obtained for different configurations.

| Parameters | Accuracy | | | |
| --- | --- | --- | --- | --- |
| | 1 cpu, 2 Gb | 1 cpu, 4 Gb | 2 cpu, 2 Gb | 2 cpu, 4 Gb |
| lr = 0.01, batch size = 20 | 0.9349 | 0.9285 | 0.9396 | 0.915 |
| lr = 0.001, batch size = 20 | 0.9491 | 0.9380 | 0.9415 | 0.9484 |
| lr = 0.0001, batch size = 20 | 0.9069 | 0.8996 | 0.9069 | 0.9038 |
| lr = 0.01, batch size = 50 | 0.9496 | 0.9546 | 0.9430 | 0.9522 |
| lr = 0.001, batch size = 50 | 0.9603 | 0.9638 | 0.9610 | 0.9585 |
| lr = 0.0001, batch size = 50 | 0.9170 | 0.9188 | 0.9206 | 0.9206 |
| lr = 0.01, batch size = 100 | 0.9616 | 0.9604 | 0.9616 | 0.9236 |
| lr = 0.001, batch size = 100 | 0.9687 | 0.9664 | 0.9679 | 0.9679 |
| lr = 0.0001, batch size = 100 | 0.9244 | 0.9059 | 0.9262 | 0.9220 |
| lr = 0.01, batch size = 150 | 0.9668 | 0.9671 | 0.9644 | 0.9667 |
| lr = 0.001, batch size = 150 | 0.9716 | 0.9703 | 0.9711 | 0.9683 |
| lr = 0.0001, batch size = 150 | 0.9296 | 0.9287 | 0.9216 | 0.9297 |

**Table 9.** License analysis of the architecture.

| Service | Docker Image | License |
| --- | --- | --- |
| Apache Airflow | apache/airflow:2.4.2 | Apache License 2.0 |
| Apache IoTDB | apache/iotdb:0.13.3 | Apache License 2.0 |
| Apache Kafka | confluentinc/cp-kafka:7.2.1 | Apache License 2.0 |
| Apache Hadoop | | Apache License 2.0 |
| Apache Horovod | | Apache License 2.0 |
| Apache Maven | | Apache License 2.0 |
| Apache MXNet | | Apache License 2.0 |
| NGINX | nginx:latest | BSD 2-clauses (d) |
| Apache NiFi | apache/nifi:1.20.0 | Apache License 2.0 |
| Apache NiFi Registry | apache/nifi-registry:1.20.0 | Apache License 2.0 |
| Apache NiFi Toolkit | apache/nifi-toolkit:1.20.0 | Apache License 2.0 |
| Postgresql | postgres:14 | PostgreSQL License (d) |
| Python | | Python Software Foundation License |
| PyTorch | | BSD 3-clauses |
| Apache Ranger | | Apache License 2.0 |
| Redis | bitnami/redis:7.0.5 | BSD 3-clauses |
| Registry | registry:2.8.1 | Apache License 2.0 |
| Apache Spark | | Apache License 2.0 |
| Apache Submarine | | Apache License 2.0 |
| Tensorflow | | Apache License 2.0 |
| Apache Zeppelin | apache/zeppelin:0.10.1 | Apache License 2.0 |
| Apache Zookeeper | bitnami/zookeeper:3.8.0 | Apache License 2.0 |

## 8. Work Limitations

In the end, the proposed architecture does not yet support important ML frameworks such as Apache Mahout (mahout.apache.org; accessed on 16 October 2022), Weka (weka.io; accessed on 16 October 2022), SparkML (spark.apache.org/docs/latest/ml-guide.html; accessed on 16 October 2022), and so forth. The described architecture is a development version but it needs adaptation to become usable in a production context as follows:

NiFi Service implements two or more instances of Apache NiFi instead of one, an NGINX proxy to load balance from port 8443 between instances of NiFi, and a NiFi Toolkit to allow the deployment of the TLS certificate. In addition, the anonymous connection allowed in the development version is replaced by user/password authentication, TLS encryption, and JKS Keystore and Truststore must be used and activated to guarantee security. Table 10 lists the composition of the NiFi Service in production.

**Table 10.** Composition of the NiFi component in production context.

| Service | Docker Image | Instance | Internal Port | Exposed Port | Mapped Port |
|---|---|---|---|---|---|
| Zookeeper | bitnami/zookeeper:3.8.0 | 1 | | 2181 | 2181 |
| NiFi | apache/nifi:1.20.0 | min 2 | | 8444, 8445 | 8444, 8445 |
| NiFi Toolkit | apache/nifi-toolkit:1.20.0 | 1 | | | |
| NiFi Registry | apache/nifi-registry:1.20.0 | 1 | | 18,000 | 18,000 |
| NGINX | nginx:latest | 1 | | 8443 | 8443 |

Kafka Service must also be adapted to improve its capabilities to accept heavy loads. Table 11 presents the three services which are deployed in the Kafka service in a production context. The instance of Zookeeper is replaced by a quorum of Zookeeper composed of three instances and the unique instance of Kafka is replaced by three instances working in parallel and coordinated by the Zookeeper quorum. Each instance of the quorum uses server ports 2888 and 3888 to communicate with other instances while each one uses separated and unique client ports: 2181, 2182, and 2183, respectively, with which each one of the tree instances of Kafka is linked. Each Kafka instance listens on two external different ports: 9092 and 29,092; 9093 and 29,093; 9094 and 29,094 respectively, and on internal ports: 19,092, 19,093, and 19,094, respectively. Kafka Schema Registry listens to requests on port 8081.

**Table 11.** Composition of the Kafka Component in production context.

| Service | Docker Image | Instance | Internal Port | Exposed Port | Mapped Port |
|---|---|---|---|---|---|
| Zookeeper | confluentinc/ cp-zookeeper:7.2.1 | 3 | 2888, 3888 | 2181,2182, 2183 | 2181, 2182, 2183 |
| Kafka | confluentinc/ cp-kafka:7.2.1 | 3 | 19,092, 19,093, 19,094 | 9092, 9093, 9094 29,092, 29,093, 29,094 | 9092, 9093, 9094 29,092, 29,093, 29,094 |
| Kafka Schema Registry | confluentinc/cp- schema-registry:7.2.1 | 1 | | 8081 | 8081 |

Apache Submarine 0.8.0 must be deployed with Kubernetes 1.21.14, Kubectl 1.21.0 (kubernetes.io/docs/reference/kubectl/kubectl/; accessed on12 February 2023), Helm 3.0.0 (helm.sh; accessed on 14 February 2023), Minikube 1.23.0 (minikube.sigs.k8s.io; accessed on 15 February 2023), istioctl 1.17.1 (istio.io; accessed on 16 February 2023), and the following dependencies: KinD 0.17 (kind.sigs.k8s.io; accessed on 16 May 2023), OpenJDK 11, Maven 3.8.0, Docker 20.10.8, NodeJS 16.19.1 LTS, Go 1.17, and Python 3.10. Afterward, the submarine-cloud-v3-system is deployed using Helm, and Kubectl is used to create the cluster. The following artifacts are also deployed: MySQL, MinIO (min.io; accessed on 27 May 2023), MLFlow (mlflow.org; accessed on 27 May 2023), and Tensorboard (tensorflow.org/tensorboard; accessed on 27 May 2023).

## 9. Conclusions and Future Directions

It is often necessary to combine disparate data sources to construct a comprehensive training database for machine learning algorithms. However, these data sources exhibit variations in terms of quality and data formats. To ensure consistency across these sources, preprocessing steps are required to consolidate the training database.This paper presents an integrated architecture associating a complete acquisition chain and a training ML platform. This architecture has been designed to combine and process data from IoT, blockchain, and open data. It also allows the complete life cycle, which enables users to deploy their models on their usual frameworks: TensorFlow, Apache MXNet, PyTorch, Keras, and Horovod.

To validate the architecture, experimentation was conducted to compare training operations with open data and blockchain data on a single CPU. Afterward, experimentation was achieved to compare the impact of the distribution of the training. The results show that the distribution of the training on the architecture does not impact significantly the accuracy, and that performances increase with a batch size of up to 150 and a learning

rate of 0.001. The architecture's ability to process and train a classification model based on data stored as files and stored within the Ethereum blockchain was evaluated. The architecture must be fully tested on complete use cases in production to definitively validate the entire architecture.

In future works, the proposed architecture for the WALLeSmart platform (wallesmart. be; accessed on 1 June 2023) [30,31], a cloud platform designed to centralize data from multiple sources from the agricultural sector in the Walloon region (Belgium), manages users' consent for the use of their data, and deploy and host applications and services for farmers, will be integrated. The proposed architecture coupled with the synthetic generation of datasets currently under development will allow for the acceleration of the development of new services and applications based on machine learning, which will then be hosted on the WALLeSmart platform (wallesmart.be; accessed on 16 February 2023). In addition, the possibility to replace Docker with Singularity (sylabs.io; accessed on 16 February 2023), which is better adapted for High-Performance Computing (HPC), will be investigated. Singularity offers a better level of security, is compatible with Docker, and allows to deploy encrypted containers [32]. Moreover, many researchers use R and R shiny for model development; therefore, it is also worth investigating the possibility of integrating this into the architecture.

**Author Contributions:** Conceptualization, O.D., J.B.N.P., M.H., A.G. and R.A.A.; methodology, R.A.A.; software, O.D. and J.B.N.P.; validation, R.A.A., K.G. and O.B.H.; formal analysis, O.D. and M.H.; investigation, O.D.; resources, A.G. and K.G.; data curation, O.B.H.; writing—original draft preparation, O.D., J.B.N.P. and M.H.; writing—review and editing, O.D.; visualization, R.A.A. and K.G.; supervision, F.L., J.B., H.S. and N.G.; project administration, M.B. and P.M.; funding acquisition, C.B. All authors have read and agreed to the published version of the manuscript.

**Funding:** This research was integrally funded Elevéo by Awé Group. Elevéo owns all intellectual property related to this research. The APC was integrally funded by MDPI Information.

**Institutional Review Board Statement:** Not applicable.

**Informed Consent Statement:** Not applicable.

**Data Availability Statement:** Not applicable.

**Acknowledgments:** The authors would like to express their gratitude to Meryem Elmoulat for their help in editing the writing of this paper.

**Conflicts of Interest:** The authors declare that they have no known competing financial interest or personal relationships that could have appeared to influence the work reported in this paper.

## Abbreviations

The following abbreviations are used in this manuscript:

| | |
|---|---|
| AGV | Automated Guided Vehicles |
| AI | Artificial Intelligence |
| AMWR | Adaptive Moving Window Regression |
| API | Application Programming Interface |
| BD | Big Data |
| BDE | Big Data Exploration |
| CEP | Complex Event Processing |
| CL | Cloud Layer |
| CoAP | Constrained Application Protocol |
| CPS | Cyber-Physical Systems |
| CPU | Central Processing Unit |
| DL | Deep Learning |
| DM | Data Mining |
| DNN | Deep Neural Network |
| DS | Data Science |
| EL | Edge Layer |

| | |
|---|---|
| FL | Fog Layer |
| GUI | Graphical User Interface |
| GPU | Graphics Processing Unit |
| HDFS | Hadoop Distributed File System |
| HILDA | Human-In-the-Loop Data Analysis |
| HKMS | Hadoop Key Management Service |
| HPC | High Performance Computing |
| HTTP | Hypertext Transfer Protocol |
| IDEAaS | Interactive Data Exploration As-a-Service |
| IoT | Internet of Things |
| IP | Internet Protocol |
| JKS | Java KeyStore |
| JSON | JavaScript Object Notation |
| KDD | Knowledge Discovery in Databases |
| LR | Learning Rate |
| LSTM | Long-Short-Term-Memory |
| ML | Machine Learning |
| MPU | Memory Protection Unit |
| MQTT | MQ Telemetry Transport |
| RKMS | Ranger Key Management Service |
| TOS | Real-Time Operating System |
| REST | Representational State Transfer |
| SGX | Software Guard Extensions |
| TCP | Transmission Control Protocol |
| TLS | Transport Layer Security |
| UI | User Interface |

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
