# Peer review of "Towards a Unified Architecture Powering Scalable Learning Models with IoT Data Streams, Blockchain, and Open Data"

_information, doi:10.3390/info14060345_

Round 1

Reviewer 1 Report (Previous Reviewer 3)

Most of the figures are too big.

Much more result data of the experiments have to be provided and analysed.

Conclusion section has to be extended.

Most of the figures are too big.

Much more result data of the experiments have to be provided and analysed.

Conclusion section has to be extended.

Author Response

Reviewer 2 Report (Previous Reviewer 2)

The article has been greatly improved. 

The motivation for the research is now clear. And, the contributions of the research work are also clearly pointed out.

The proposed architecture conceptualization is very complete, providing a sound components overview and a thorough explanation of its components/layers.

The Experimentation / Validation of the proposal has been greatly improved. And, the limitations of the proposal are clearly stated.

English language needs improvement.

English language needs improvement.

Author Response

This manuscript is a resubmission of an earlier submission. The following is a list of the peer review reports and author responses from that submission.

Round 1

Reviewer 1 Report

- In the related works section, the authors presented some research works and some platforms!! It was wise to describe only work related to this issue. And another section to describe the different platforms. Still it was preferable to present a comparative table between platforms.

- The problem statement intended by the authors needs to be further described;

- The authors mention in their text that the data is validated and verified! how ?

- This article lacks a conceptual approach! no UML diagram was presented; no algorithm!!

- No complexity study of this present work has been small and discussed!!

- The interpretation of the results obtained requires more discussion and criticism!

Reviewer 2 Report

The article addresses an important issue, the one of dealing with the massive amounts of data produced by IoT, when wanting to use that data for training machine learning models.

The introduction is very short. A longer motivational argumentation is needed, along with examples/cases of problematic use of IoT data for ML.

Figure 2 needs a reference, if it has been taken from elsewhere, or it needs further explanation/justification, if it is created by the authors.

The Related works section is very short and incomplete, also needing further development and improvement.

In the architectural proposal section, a table with the technologies/frameworks/tools used, and the role each one plays in the architecture proposed, is lacking. This table would help in reading the presented text.

A more developed explanation of each architectural layer would also help in better understanding the proposal.

In the Implementation section, Fig 5 does increase confusion, or maybe more explaining text is lacking. In Fig. 5, one cannot understand the frase "Horovod is  a distributed deep-learning training framework that can be used with TensorFlow, Keras, PyTorch, and Apache MXNet.", as Keras isn't represented, and Horovod appears as one of the ML platforms/APIs.

In fact, sections 4, 5 and 6 (Implementation, Experimentation, and Results and discussion) need further development so that a reader can understand the scope of the project and all its problems and benefits.

The Conclusions also need to be better grounded in the results of experimentation and discussion.

Some sentences need to be rewritten for improving English.

Reviewer 3 Report

I can't find the reserach contribution of the manuscript. Morover, it has a lot of drawbacks, e.g.
  1. "The traditional approach consists to develop specific mathematical or statistical models for an application domain." - what approches? what domains?  what do you mean?
  2. what if the contributon of figure 1?
  3. specify clearly the contribution of the manuscript
  4. "In section 2, we conduct a survey and summarize recent advances in ML platforms. " - which survey???
  5. "In section 3, we conceptualize and describe our architectural proposal. " - "proposal" of what?
  6. "In section 4, we implement our architecture and achieve experimentation to demonstrate its abilities and performances. " - you don't implement anything in he section - you can present e.g. the implementation process, its results, ...
  7. related works section has to be extended
  8. why you proposed architecture is for "IoT Data Streams, Blockchain, and Open Data" ? it is not presented, explained and justifid
  9. the layers of the propoesed architecture have to be presented in details
  10. section 5 has to be extended
  11. section 6 has to be extended
  12. conlusions section has to be extended
  13. English should be improved
  14. write your manuscript in impersonal form
  15. bibliogrpahy has to be extended

English should be improved
